# Road traffic accident-related thoracic trauma: Epidemiology, injury pattern, outcome, and impact on mortality—A multicenter observational study

**Axel Benhamed**[1,2,3,4⊚¤]*, **Amina Ndiaye**[5‡], **Marcel Emond**[3,4‡], **Thomas Lieutaud**[5‡], **Valérie Boucher**[4‡], **Amaury Gossiome**[1], **Bernard Laumon**[5], **Blandine Gadegbeku**[5‡], **Karim Tazarourte**[1,2⊚]

**1** Service d'Accueil des Urgences–SAMU 69, Centre Hospitalier Universitaire Édouard Herriot, Lyon, Hospices Civils de Lyon, France, **2** INSERM U1290 (RESHAPE), Université de Lyon 1, Lyon, France, **3** Département d'urgences, Centre Hospitalier Universitaire de Québec-Université Laval, Québec, Québec, Canada, **4** Research Centre, CHU de Québec-Université Laval, Québec, Québec, Canada, **5** IFSTTAR, Université Gustave Eiffel, Bron, France

⊚ These authors contributed equally to this work.
¤ Current address: Hospices Civils de Lyon, Service d'accueil des Urgences–SAMU 69, Centre Hospitalier Universitaire Édouard Herriot, Lyon, France
‡ AN, ME, TL, VB, BG also contributed equally to this work.
* axel.benhamed@chu-lyon.fr

**Data Availability Statement:** All relevant data are within the paper.

## Abstract

### Background

Thoracic trauma is a major cause of death in trauma patients and road traffic accident (RTA)-related thoracic injuries have different characteristics than those with non-RTA related thoracic traumas, but this have been poorly described. The main objective was to investigate the epidemiology, injury pattern and outcome of patients suffering a significant RTA-related thoracic injury. Secondary objective was to investigate the influence of serious thoracic injuries on mortality, compared to other serious injuries.

### Methods

We performed a multicenter observational study including patients of the Rhône RTA registry between 1997 and 2016 sustaining a moderate to lethal (Abbreviated Injury Scale, AIS≥2) injury in any body region. A subgroup (AIS$_{Thorax}$≥2 group) included those with one or more AIS≥2 thoracic injury. Descriptive statistics were performed for the main outcome and a multivariate logistic regression was computed for our secondary outcome.

### Results

A total of 176,346 patients were included in the registry and 6,382 (3.6%) sustained a thoracic injury. Among those, median age [IQR] was 41 [25–58] years, and 68.9% were male. The highest incidence of thoracic injuries in female patients was in the 70–79 years age group, while this was observed in the 20–29 years age group among males. Most patients

**Funding:** The Rhône registry is supported by the Gustave Eiffel university, the French public health agency and the road safety directorate (ministry of the interior). The authors declare that the funding sources had no role in the conduct, analysis, interpretation, or writing of this manuscript.

**Competing interests:** The authors declare that they have no competing interests.

were car occupants (52.3%). Chest wall injuries were the most frequent thoracic injuries (62.1%), 52.4% of which were multiple rib fractures. Trauma brain injuries (TBI) were the most frequent concomitant injuries (29.1%). The frequency of $MAIS_{Thorax} = 2$ injuries increased with age while that of $MAIS_{Thorax} = 3$ injuries decreased. A total of 16.2% patients died. Serious (AIS$\geq$3) thoracic injuries (OR = 12.4, 95%CI [8.6;18.0]) were strongly associated with mortality but less than were TBI (OR = 27.9, 95%CI [21.3;36.7]).

## Conclusion

Moderate to lethal RTA-related thoracic injuries were rare. Multiple ribs fractures, pulmonary contusions, and sternal fractures were the most frequent anatomical injuries. The incidence, injury pattern and mechanisms greatly vary across age groups.

## Introduction

In the year 2000, road traffic accidents (RTA) were the tenth leading cause of death in the world, the eighth in 2016, and could become the fifth by 2030 according to the World Health Organization [1, 2]. Victims of RTA often suffer multi-trauma and present with thoracic injuries in about 50% cases [3–5]. Furthermore, thoracic trauma are the third most common cause of death in multi-trauma patients [6] and are associated with poor short-term outcomes as they are responsible for up to 25% of trauma-related deaths [7, 8]. Therefore, thoracic traumas represent a major medical and economic problem, and providing care for those patients is challenging. Some risk factors of mortality following thoracic trauma have already been identified, including of the presence of specific structural damages to the chest wall and thoracic organs [8–10] and many studies have shown that thoracic injuries significantly contribute to the mortality of multi-trauma patients, in adults as well as in the paediatric population [11–13]. Furthermore, recent literature has shown that patients with RTA-related thoracic injuries had different clinical characteristics than those with non-RTA related thoracic traumas [5]. However, the literature on the epidemiology, injury pattern and outcome of patients with thoracic trauma following a road traffic accident is still scant.

The main objective of this study was to investigate the epidemiology, injury pattern and outcome of patients suffering one or more RTA-related thoracic injury. Secondary objective was to investigate the influence of serious thoracic injuries on traumatic mortality compared to other types of serious injuries.

## Methods

### Study design and setting

We conducted a retrospective study that included patients who sustained a RTA between January 1997 and December 2016.

We used prospectively recorded data from the Rhône RTA registry (*Registre des victimes d'accidents de la circulation du Rhône*), which was implemented in 1995. This registry covers the Rhône area of France (1.83 million inhabitants, 676 inhabitants/km$^2$) and its use is approved by the relevant national authorities (*Comité National des Registres*, CNR) and data protection agency and (*Commission Nationale de l'Informatique et des Libertés*, CNIL; N° 999211).

The registry includes the demographic characteristics of each RTA casualty and a description of the sustained body injuries. Patient information is collected prospectively during three consecutive time periods from the accident site to hospital discharge: prehospital scene, emergency room or intensive care unit (ICU), and discharge. The registry team supervises the verification of the data from different sources about the same accident or victim, coding, storage and filing, and statistical analysis. Each injury is coded according to the Abbreviated Injury Scale, a severity score, ranging from 1 (minor) to 6 (beyond treatment) [14]; between 1996 and 2014, the 1990 AIS version was used and the 2005 AIS version was used for 2015 and 2016. The injury severity score (ISS) is calculated from the three worst-affected body regions as the sum of squares of the respective AIS severity levels. The full data collection method has been previously described [15].

## Study population

Patients are included in the registry if they sustained a RTA involving at least one vehicle (motorised or not) in the Rhône area, which required institutional healthcare from one of the 245 private and public healthcare structures (including level I, II and III trauma centers) cooperating together, including prehospital primary care teams and forensic medicine institutes.

All patients included in the registry with one or more AIS$\geq$2 injury in any body region (AIS$\geq$2 group) were considered in our analyses.

## Measures

Several variables were extracted and analysed: patient characteristics (sex, age, road user category), road network, anatomical injuries, severity score (AIS and ISS), and outcome (ICU admission and mortality). As victims could suffer from several injuries in each body region, the maximum AIS (MAIS) was scored using the injury of the highest severity.

## Main outcome measure

The main outcome of our study is RTA-related thoracic injury. We have defined a subgroup of patients in the registry (AIS$_{Thorax}\geq$2), which included all patients with one or more thoracic injury (AIS$\geq$2).

## Secondary outcome measure

Death, our secondary outcome, is medically certified either at the scene or noted in medical charts during hospital stay. An autopsy is systematically undertaken in patients dying in the prehospital setting to confirm cause of death and to provide a complete injury assessment based on AIS scoring.

## Ethics approval and consent to participate

The registry has been approved by the relevant French authority and national data protection commission (*Comité National des Registres*, CNR, and *Commission Nationale de l'Informatique et des Libertés*, CNIL, N˚ 999211). Patients, or parents/guardians received information about inclusion in the registry but the need for consent was waived. All data were fully anonymized before analysis.

## Statistical analysis

Baseline characteristics were described by frequencies and percentages for categorical variables, and medians and interquartile range [IQR] for continuous variables. We compared the

groups using the Pearson $Chi^2$ test for categorical variables. We performed a multivariate logistic regression based on complete cases for our secondary outcome (mortality). The model was built using the following covariates: age, sex, global severity (ISS), road user, road network, year of inclusion. Significant prognostic variables at 5% significance on the univariate analysis were included in analysis. The odds ratio (OR) for each risk factor investigated was calculated as well as the corresponding 95% confidence intervals (CI). Missing data were not imputed Statistical analyses were performed using SAS (Statistical Analysis System v9.4, SAS Institute Inc., Cary, NC, USA). In all analyses, p<0.05 was considered as statistically significant.

## Results

### Patient and injury characteristics

Over the study period, a total of 176,346 RTA victims were included in the registry; 46,526 (26.4%) had at least one moderate to lethal injury (AIS≥2 group) and 6,382 (3.6%) had at least one thoracic injury $AIS_{Thorax}$≥2 ($AIS_{Thorax}$≥2 group). Between the first period (1997–2001) to the last one (2012–2016), we noted a 38.7% decrease of patients included in the AIS≥2 group while this was of 23.6% in the group of patients with a thoracic trauma.

In the $AIS_{Thorax}$≥2 group, the median [IQR] age was 41 years [25–58], and 4,400 (68.9%) patients were male. Most of patients were car occupants (52.3%, n = 3,337) and motorcyclists (25.3%, n = 1,617). In most cases (52.7%), the RTA occurred on a city street (Table 1). The highest incidence of $AIS_{Thorax}$≥2 injuries in female patients was in the 70–79 years age group (23.1/100,000 inhabitants), while this was observed in the 20–29 years age group among males (39.1/100,000 inhabitants; Fig 1). The distribution of road user by age group is reported in the Fig 2.

### Injury pattern

A total of 8,729 thoracic injuries were reported (some patients may have sustained multiple thoracic injuries). Of these, chest wall injuries were the most frequent (62.1%, n = 5,419), over half of which were multiple rib fractures (52.4%). Lung injuries were the second most frequent type of thoracic injuries (24.7%, n = 2,158), 88.7% of which were lung contusions. Pleural injuries (including pneumothorax and haemothorax) were found in 5.3% (n = 466) of cases (Table 2). The frequency of $MAIS_{Thorax}$ = 2 injuries increased with age while that of $MAIS_{Thorax}$ = 3 injuries decreased (Fig 3). The most frequent concomitant AIS≥2 extra-thoracic injuries affected the head (29.1%), upper extremities (26.8%) and lower extremities (25.8%) The head (19.9%) was the most serious (AIS≥3) concomitant body area injured (Table 3).

### Severity and mortality

The median ISS [IQR]was 14 [6–27]. A total of 30.8% (n = 1,968) thoracic trauma patients were admitted to an ICU, and 16.2% (n = 1,031) patients died (respectively 17.4%, 16.6%, 16.8%, 13.5% of included patients by time-period as described in Table 1). Of those, 61.5% (n = 634) died on-scene. Among those admitted to hospital (n = 397), 37.3% (n = 148) died during the first 24 hours, and 75.6% (n = 300) died within the first three days. Thoracic injuries were the cause of death for 42.7% (n = 440) of our 1,031 deceased patients. Pedestrians had the highest mortality proportion (30.6%, n = 207), followed by motorcyclists (17.6%, n = 285), car occupants (13.6%, n = 455) and cyclists (10.5%, n = 46). A total of 1.4% (n = 36) $MAIS_{Thorax}$ = 2 patients died while 7.9% (n = 164) with $MAIS_{Thorax}$ = 3 died, and 47.3% (n = 831/1757) patients with $MAIS_{Thorax}$ ≥4 died.

**Table 1. Demographics and mechanism.**

| | AIS≥2 group | AIS_Thorax≥2 group |
|---|---|---|
| | (n = 46,526) | (n = 6,382) |
| | n (%) | n (%) |
| Year of inclusion | | |
| 1997–2001 | 14,771 (31.7) | 1,896 (29.7) |
| 2002–2006 | 12,249 (26.3) | 1,576 (24.7) |
| 2007–2011 | 10,458 (22.5) | 1,462 (22.9) |
| 2012–2016 | 9,048 (19.5) | 1,448 (22.7) |
| Age[a], years median [IQR] | 28 [18–45] | 41 [25–58] |
| Sex[b] | | |
| Male | 33,738 (72.5) | 4,400 (68.9) |
| Female | 12,778 (27.4) | 1,982 (31.1) |
| Road user | | |
| Car occupant | 12,739 (27.4) | 3,337 (52.3) |
| Pedestrian | 6,011 (12.9) | 677 (10.6) |
| Bicyclist | 8,783 (18.9) | 439 (6.9) |
| Motorcyclist | 14,497 (31.2) | 1,617 (25.3) |
| Other | 4,496 (9.6) | 312 (4.9) |
| Road network | | |
| City street | 26,238 (56.4) | 3,361 (52.7) |
| Highway | 3,006 (6.5) | 723 (11.3) |
| Rural road | 5,745 (12.3) | 1,497 (23.4) |
| Other | 11,537 (24.8) | 801 (12.6) |

[a]45 missing data

[b]10 missing data

AIS ≥2 group: trauma patients presenting with at least one injury AIS≥2.

AIS_Thorax≥2 group: trauma patients presenting with at least one thoracic injury AIS≥2.

AIS: abbreviated injury scale.

## Impact of thoracic trauma on mortality and other factors associated with mortality

We investigated the influence of serious (AIS ≥ 3) thoracic injuries on traumatic mortality compared to other serious injuries. Our multivariate analysis revealed that patients with serious traumatic brain injuries were associated with a higher risk of death (OR = 27.9, 95%CI [21.3–36.7]) than those with serious thoracic injuries (OR = 12.4, 95%CI [8.6–18.0]).

Other factors associated with mortality in the AIS≥2 group were all age groups above 39 years compared to 20–39 years group, male sex (OR = 1.4, 95%CI [1.2–1.7]), RTA occurring in a highway (OR = 1.8, 95%CI [1.5–2.3]) or a rural road (OR = 1.8, 95%CI [1.5–2.1]) compared to those in a city street (Table 4).

## Discussion

The results of our study show that moderate to lethal RTA-related thoracic injuries were surprisingly rare, affecting less than four out of every 100 patients, and most frequently occurred in middle-aged male patients.

We have found a lower incidence of thoracic trauma compared to the numbers previously reported [10, 16, 17]. Peek *et al.* recently reported that single rib fracture accounted for 20% of

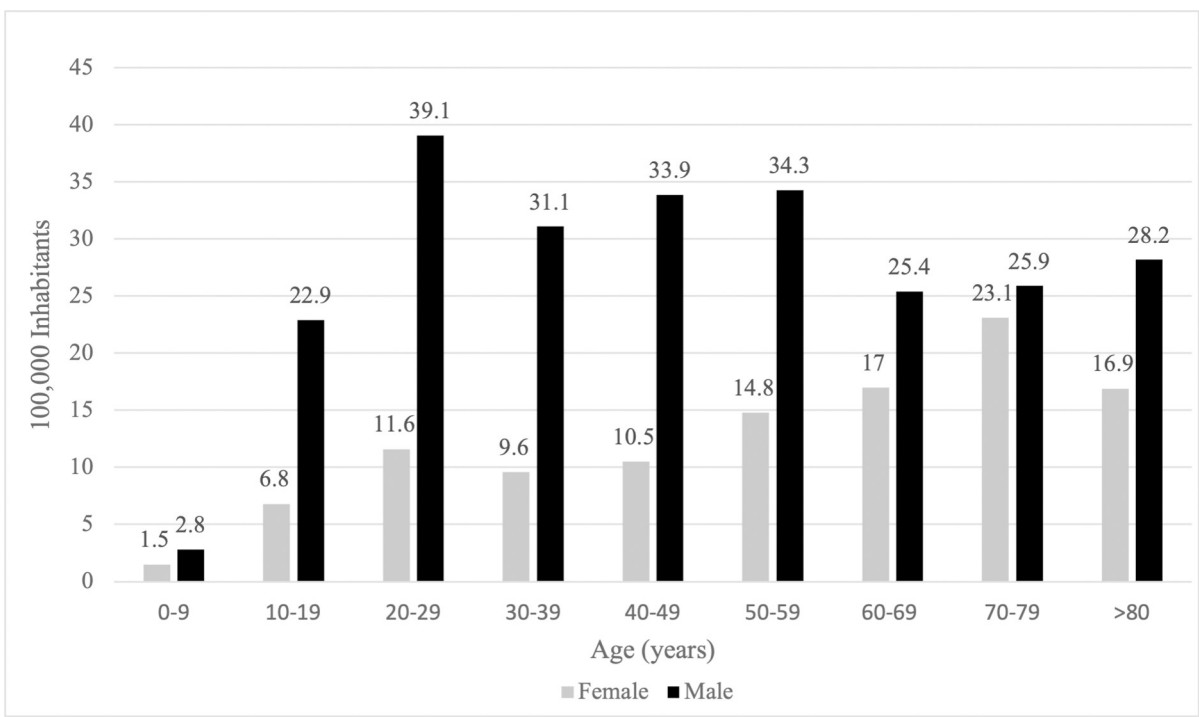

**Fig 1. Road traffic accident among the AIS$_{Thorax}\geq$2 group per 100,000 inhabitants in the Rhône area population.** Subgroups according to the age. AIS$_{Thorax}\geq$2 group: trauma patients presenting with at least one thoracic injury AIS$\geq$2.AIS: abbreviated injury scale.

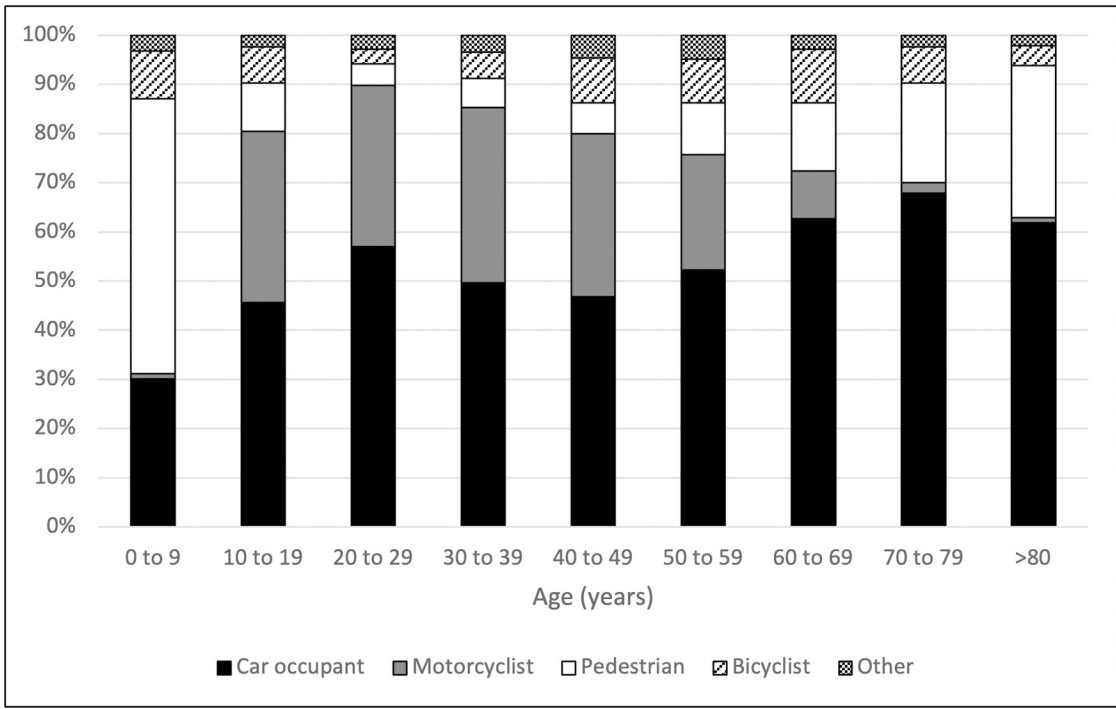

**Fig 2. Road user category distribution among the AIS$_{Thorax}\geq$2 group.** Subgroups according to the age. AIS$_{Thorax}\geq$2 group: trauma patients presenting with at least one thoracic injury AIS$\geq$2. AIS: abbreviated injury scale. MAIS: maximum abbreviated injury scale.

**Table 2. Description of thoracic injuries in the $AIS_{Thorax} \geq 2$ group.**

| Injury characteristics[a] | n (%) |
|---|---|
| Chest wall injuries | 5,419/8,729 (62.1) |
| Multiple rib fracture | 2,842/5,419 (52.4) |
| Sternal fracture | 1,719/5,419 (31.7) |
| Flail chest | 478/5,419 (8.8) |
| Single rib fracture | 276/5,419 (5.1) |
| Other | 104/5,419 (2) |
| Lung injuries | 2,158/8,729 (24.7) |
| Pulmonary contusion | 1,914/2,158 (88.7) |
| Pulmonary laceration | 244/2,158 (11.3) |
| Pleural and mediastinal injuries | 466/8,729 (5.3) |
| Pneumo and/or haemothorax | 352/466 (75.5) |
| Pneumo and/or haemomediastinum | 105/466 (22.5) |
| Other | 9/466 (2) |
| Cardiac or vascular injuries | 421/8,729 (4.8) |
| Cardiac | 227/421 (54) |
| Thoracic aorta | 120/421 (28.5) |
| Pulmonary arteries/veins | 46/421 (11) |
| Coronary artery | 12/421 (2.9) |
| Subclavian artery/vein | 10/421 (2.4) |
| Other | 5/421 (1.2) |
| Other injuries | 265/8,729 (3) |
| Skin injuries | 151/265 (57) |
| Diaphragmatic injuries | 83/265 (31.3) |
| Tracheal and bronchial injuries | 21/265 (7.9) |
| Oesophageal injuries | 10/265 (3.8) |

[a]No missing data

One patient could have suffered from multiple thoracic injuries, therefore the total of injuries (n = 8,729) presented in the table is greater than the number of $AIS_{Thorax} \geq 2$ patients (n = 6,382).

$AIS_{Thorax} \geq 2$ group: trauma patients presenting with at least one thoracic injury AIS$\geq$2.

AIS: abbreviated injury scale.

rib fracture cases [18, 19]. This difference may be explained by the fact that we chose not to include these patients. Indeed, we did not include minor injuries because previous studies have demonstrated that these single rib fractures were not associated with mortality [10]. Another explanation could be that the present study may suffer a measurement bias in the late 90's and early 2000s because some patient may have undergone a traditional selected CT scan that did not explore the thorax. The recent democratization of whole body computed tomography (WBCT) which became standard practice in many centers over the world in the last two decades resulted in more diagnosed injuries [20]. Another interesting finding of the present study is the evolution of $AIS_{Thorax}$ 2 and 3 with age. Children aged between 0 and 9 years had the highest incidence of AIS 3 injuries, and this decreased with age. This finding is similar to Samarasekera *et al.*'s, as they reported a very high incidence of AIS>2 thoracic injuries (88%) among children aged under 15years, 65% of which being lung contusions. The more compressible and incompletely calcified thoracic skeleton of children, which allows the transmission of large forces to the thoracic cavity structures, making rib fractures uncommon, may partially explain this. Thus, high-energy impact trauma may cause major internal injuries with

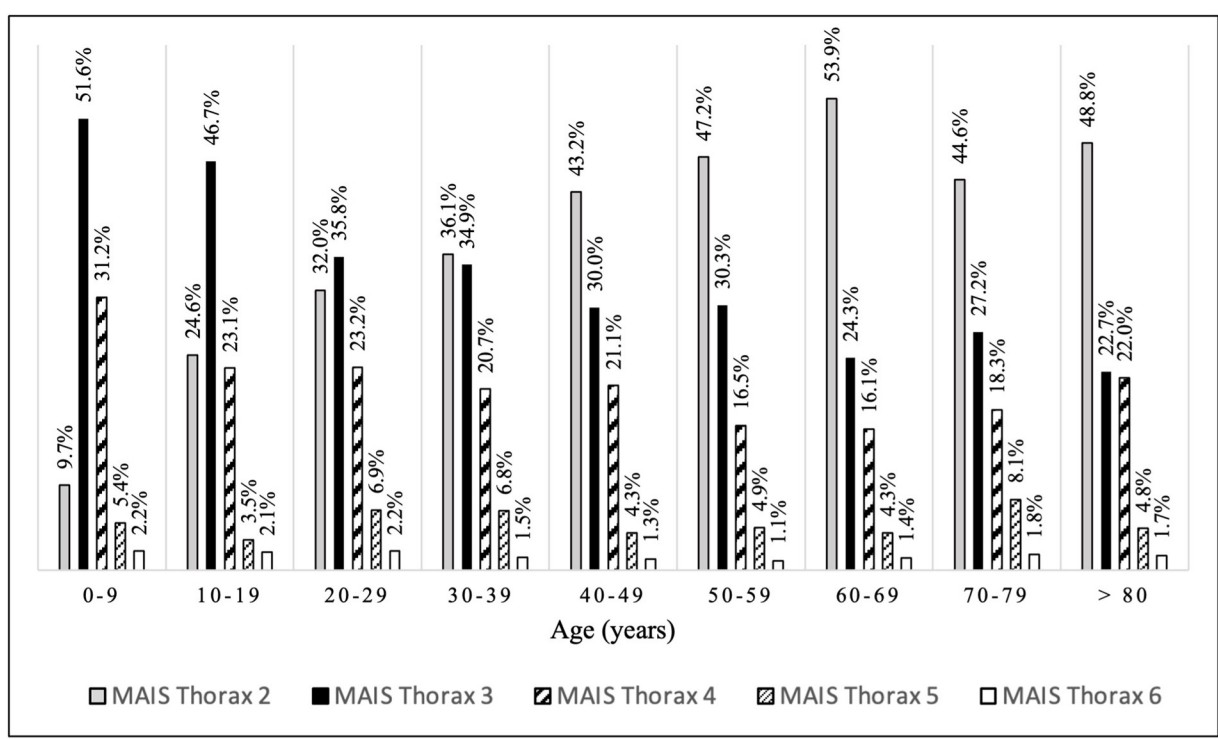

**Fig 3. Thoracic MAIS distribution among the AIS$_{Thorax}\geq$2 group.** Subgroups according to the age. AIS$_{Thorax}\geq$2 group: trauma patients presenting with at least one thoracic injury AIS$\geq$2. AIS: abbreviated injury scale. MAIS: maximum abbreviated injury scale.

little evidence of external injuries or fractures of the bony thorax as reported by many authors [12, 21, 22]. Furthermore, since children are more frequently involved in RTA as pedestrians, they may be at higher risk of projection and severe trauma as a result of a direct impact with the front bumper because of their height. In contrast, the incidence of moderate (AIS = 2) thoracic injuries increased with age. These injuries are mainly rib fractures. which, in older patients, were reported to be a frequent consequence of a RTA [23]. With muscle atrophy and osteoporosis, which are commonly associated with older age, less force may be required to cause rib fractures in this population. Herein, mortality was similar to that reported by several authors [7, 10, 24]. Other studies showed lower mortality rates (5% and 5.5%) [13, 25] but these only included patients reaching the hospital with vital signs, whereas on-scene dead patients were not excluded from our analyses. Had these patients been excluded, our mortality rate would have been 6.2%. Similarly to other authors [10], we have found serious thoracic trauma were also strongly associated with patient mortality, as three-quarters serious thoracic trauma-related deaths occurred within the first 24 hours. This contradicts Grubmüller *et al.*'s results, which found no association between the presence of serious thoracic trauma and mortality [24]. This may be explained by the potential limited external validity of their results, as their study included patients from a single high-volume level-I trauma center. It was indeed suggested in the literature that the hospital volume of severely injured patients may be an independent predictor of survival [26, 27]. In addition to the fact that our study included data from multiple centers with different levels of trauma care designation, their population is different than ours as they excluded patients with penetrating trauma and those in whom resuscitation efforts outside the hospital have failed.

**Table 3. Concomitant extra-thoracic injuries between $AIS_{Thorax} \geq 2$ group and $AIS_{Thorax} < 2$ group.**

| Body region[a] | AIS≥2 group | $AIS_{Thorax} \geq 2$ group |
|---|---|---|
| | (n = 46,526) | (n = 6,382) |
| **Head** | n (%) | n (%) |
| AIS≥2 | 12,096 (26) | 1,855 (29.1) |
| AIS≥3 | 3,162 (6.8) | 1,268 (19.9) |
| **Face** | | |
| AIS≥2 | 2,181 (4.7) | 477 (7.5) |
| AIS≥3 | 186 (0.4) | 93 (1.5) |
| **Neck** | | |
| AIS≥2 | 132 (0.3) | 68 (1.1) |
| AIS≥3 | 59 (0.1) | 33 (0.3) |
| **Abdomen/pelvis** | | |
| AIS≥2 | 2,087 (4.5) | 1,143 (17.9) |
| AIS≥3 | 1,005 (2.2) | 604 (9.5) |
| **Spine** | | |
| AIS≥2 | 3,438 (7.4) | 1039 (16.3) |
| AIS≥3 | 731 (1.6) | 322 (5.0) |
| **Upper extremity** | | |
| AIS≥2 | 19,038 (40.9) | 1,713 (26.8) |
| AIS≥3 | 3,363 (7.2) | 454 (7.1) |
| **Lower extremity** | | |
| AIS≥2 | 15,198 (32.7) | 1,646 (25.8) |
| AIS≥3 | 5,032 (10.8) | 987 (15.5) |
| **Skin** | | |
| AIS≥2 | 69 (0.1) | 5 (0.08) |
| AIS≥3 | 42 (0.09) | 2 (0.03) |

[a]No missing data

AIS≥2 group: trauma patients presenting with at least one injury AIS≥2.

$AIS_{Thorax} \geq 2$ group: trauma patients presenting with at least one thoracic injury AIS≥2.

AIS: abbreviated injury scale.

One major strength of the present study was the inclusion of on-scene deceased patients without inducing any misclassification bias because an autopsy was systematically undertaken in those patients to confirm the cause of death and provide a complete injury assessment based on AIS scoring. Furthermore, our large multicenters cohort which includes consecutive patients from the past 20 years is a non-negligeable strength.

This study has several limitations, including the update of AIS during the study period [28]. Nevertheless, we chose to base severity assessment on the widely used AIS because it is an accurate, objective and validated method to independently evaluate the impact of each body region on mortality [29]. Besides, Hsu *et al.* [30] found that the AIS update had no impact on mean ISS when considering the thoracic body region. Furthermore, our retrospective study design, with its known weaknesses, may be considered a limit. However, the present study is based on prospectively collected multicenters data over a 20-year-period including level I to III trauma centers and included patients of all ages whereas many studies focused on thoracic trauma are single center studies and only included patients ≥16 years [7, 9, 13, 16, 24, 31]. Also, our secondary outcome relies on an anatomic system of injury classification (AIS and ISS) since the database does not report physiological assessment nor medical management.

**Table 4. Predictors of mortality in multivariate analysis in AIS≥2 group.**

| | OR [95%CI] |
|---|---|
| **Body region**[a] | |
| Head | 27.9 [21.3–36.7] |
| Face | 11.5 [7.3–18.2] |
| Neck | 2.4 [0.8–6.8] |
| Thorax | 12.4 [8.6–18.0] |
| Abdomen and pelvis | 10.8 [7.5–15.4] |
| Spine | 3.4 [2.4–4.7] |
| Upper extremity | 2.7 [2.1–3.5] |
| Lower extremity | 5.2 [4.1–6.5] |
| **Age, years** | |
| 0–9 | 0.5 [0.3–0.8] |
| 10–19 | 0.7 [0.5–0.8] |
| 20–39 | 1 |
| 40–59 | 1.3 [1.1–1.6] |
| 60–79 | 2.1 [1.7–2.6] |
| ≥ 80 | 6.1 [4.6–8.0] |
| **Sex, male** | 1.4 [1.2–1.7] |
| **Road user** | |
| Car occupants | 1 |
| Pedestrians | 1.1 [0.9–1.4] |
| Bicyclists | 0.4 [0.3–0.6] |
| Motorcyclists | 0.8 [0.7–1.0] |
| Other | 0.8 [0.6–1.1] |
| **Road network** | |
| City street | 1 |
| Highway | 1.8 [1.5–2.3] |
| Rural road | 1.8 [1.5–2.1] |
| Other | 0.5 [0.4–0.6] |
| Year of inclusion | 0.95 [0.94–0.96] |

[a] Impact of a MAIS ≥3 injury compared to a MAIS = 2 injury in the same body region

AIS≥2 group: trauma patients presenting with at least one injury AIS≥2, MAIS: Maximum abbreviated injury scale.

p<0.001 for all variables.

Therefore, there is no data regarding prehospital management, time to surgery, ICU management such as airway and ventilation management, use of vasopressors or massive blood transfusion, and no detail on organ and respiratory failure or pre-existing chronic comorbid diseases which have been reported to influence outcome in thoracic trauma patients [8–10, 24]. The value of anatomic scoring systems in outcome prediction of trauma patients has been debated and compared to physiological scores. Some authors found the ISS to be a better severity predictor than the revised Trauma Scale (RTS) and the simplified acute physiology scale II (SAPS II) [32]. Hence, we believe that using anatomic scoring in the present study is as efficient as physiological scoring in predicting outcomes. At last, we included patients prior to the adoption of WBCT which may have contributed to underreporting of some injuries such as pulmonary contusions and rib fractures.

## Conclusion

Moderate to lethal RTA-related thoracic injuries were rare. Multiple ribs fractures, pulmonary contusions, and sternal fractures were the most frequent anatomical injuries. The incidence, injury pattern and mechanisms greatly vary across age groups.

## Acknowledgments

The authors would like to thank Philip Robinson for his linguistic advice with this manuscript and everyone who participated in the data collection and data recording for the Rhône Road Accident Registry Association (ARVAC: president, E. Javouhey), and University Gustave Eiffel–campus of Lyon–TS2-Umrestte (scientific coordinator, B. Laumon, medical coordinator, A. Ndiaye and administrative coordinator, B. Gadegbeku).

## Author Contributions

**Conceptualization:** Axel Benhamed, Amina Ndiaye, Karim Tazarourte.

**Formal analysis:** Amina Ndiaye, Blandine Gadegbeku.

**Funding acquisition:** Amina Ndiaye, Thomas Lieutaud, Bernard Laumon, Blandine Gadegbeku.

**Investigation:** Axel Benhamed, Amina Ndiaye, Karim Tazarourte.

**Methodology:** Axel Benhamed, Amina Ndiaye, Karim Tazarourte.

**Project administration:** Bernard Laumon.

**Supervision:** Karim Tazarourte.

**Validation:** Axel Benhamed, Amina Ndiaye, Karim Tazarourte.

**Visualization:** Axel Benhamed.

**Writing – original draft:** Axel Benhamed, Marcel Emond, Valérie Boucher, Karim Tazarourte.

**Writing – review & editing:** Axel Benhamed, Amina Ndiaye, Marcel Emond, Thomas Lieutaud, Valérie Boucher, Amaury Gossiome, Bernard Laumon, Blandine Gadegbeku, Karim Tazarourte.

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
