## [Decision Letter · Decision Letter 0]

24 Feb 2022

PONE-D-22-02469Road traffic accident-related thoracic trauma: epidemiology, injury pattern, outcome, and impact on mortality – a multicenter observational study from 1997 to 2016PLOS ONE

Dear Dr. BENHAMED,

Thank you for submitting your manuscript to PLOS ONE. After careful consideration, we feel that it has merit but does not fully meet PLOS ONE’s publication criteria as it currently stands. Therefore, we invite you to submit a revised version of the manuscript that addresses the points raised during the review process.

Both reviewers see substantial merit in your manuscript. They have provided important comments for your consideration with both providing suggestions for managing the long timeframe of the underlying dataset and how best to report on the data. Please do give the reviewers' comments careful consideration.

We look forward to receiving your revised manuscript.

Kind regards,

Belinda J Gabbe, PhD

Academic Editor

PLOS ONE

Journal Requirements:

Reviewers' comments:

Reviewer's Responses to Questions

**Comments to the Author**

1. Is the manuscript technically sound, and do the data support the conclusions?

Reviewer #1: Yes

Reviewer #2: Partly

2. Has the statistical analysis been performed appropriately and rigorously? 

Reviewer #1: Yes

Reviewer #2: Yes

3. Have the authors made all data underlying the findings in their manuscript fully available?

Reviewer #1: Yes

Reviewer #2: Yes

4. Is the manuscript presented in an intelligible fashion and written in standard English?

Reviewer #1: Yes

Reviewer #2: Yes

5. Review Comments to the Author

Reviewer #1: Thank you for the opportunity to read and comment on this interesting manuscript by Benhamed, et al., entitled, “Road traffic accident-related thoracic trauma: epidemiology, injury pattern, outcome, and impact on mortality – a multicenter observational study from 1997 to 2016.” This is a retrospective, cohort study using data from the Rhone RTA registry, aimed at describing patient characteristics, injury patterns, and outcome among patients with one or more thoracic injuries sustained in an RTA during a 20-year interval. They find that RTA-related thoracic injuries were rare (3.6%) and injury patterns and mechanisms varied by age group.

1. The manuscript is organized and the prose is clearly written. Overall, the authors have done a fine job of describing the cohort and identifying trends in the data, particularly with regard to age. However, I think much more can be accomplished with the data they already have in hand.

The authors need to take the element of time into consideration given the 20-year span of the dataset. Trauma care has changed substantially over the 20 years of the study. At the very least, year of injury could be included in the regression models as random parameters. One could go so far as comparing standardized survival ratios over time. However, this is completely at the authors’ discretion. Nonetheless, the matter of time needs to be acknowledged as a variable in Table 1 and included in the results, as appropriate.

2. The authors give a good description of the limitations of the Rhone RTA registry. Is there information regarding chronic comorbid diseases in the Rhone RTA registry? If such data are available, inclusion in the analyses would be instructive. If not, fine.

3. How was missing data handled? Was there any missing data? Some mention of missingness would be useful to readers.

4. The description of the logistic model fitting process is appreciated. Could the authors please include the area under the receiver operating characteristic curve to convey the discrimination of the model?

5. The figures are clearly presented and are very instructive. Well done.

Minor point:

1. The third sentence of the introduction, line 80, “Furthermore…” needs to be clarified. The phrases “third most common cause of death…” and “up to 25% of trauma-related deaths” need to be reconciled somehow.

Thank you for your kind consideration of my comments and questions. I look forward to your replies and revisions.

Reviewer #2: Thank you for the opportunity to read this well written, descriptive analysis of road crash related thoracic trauma from the Rhone region.

However, the authors have indicated 2 significant flaws in the paper.

The first is the inclusion of subjects prior to the adoption of whole body CT scanning. The diagnoses of rib fractures and pulmonary injuries - for example - can be considered under-reported.

Secondly , road crash fatalities rates had reduced by 50% in developed systems during the reporting period (1997-2018), yet the cohort is presented as a uniform population (see, for example, https://www.statista.com/statistics/437904/number-of-road-deaths-in-france/ accessed 21Feb2022).

Both these factors make interpreting the data difficult.

Would the authors consider:

1. Including more contemporaneous data (e.g. 1997-2021)?

2. Analysing the incidence of diagnoses as well as the changes in mortality and outcomes over time?

6. PLOS authors have the option to publish the peer review history of their article (what does this mean?). If published, this will include your full peer review and any attached files.

Reviewer #1: **Yes: **Alan Cook, MD, MS, FACS

Reviewer #2: **Yes: **Mark C Fitzgerald

---

## [Author Response · Author response to Decision Letter 0]

1 Apr 2022

We would like to truly thank all the reviewers for their helpful comments and suggestions, which we believe will greatly improve the quality of our work. 

Please find attached with the revision documents a point-by-point response to it.

Thank you very much,

Best regards,

---

## [Decision Letter · Decision Letter 1]

19 Apr 2022

PONE-D-22-02469R1Road traffic accident-related thoracic trauma: epidemiology, injury pattern, outcome, and impact on mortality – a multicenter observational studyPLOS ONE

Dear Dr. BENHAMED,

Thank you for submitting your manuscript to PLOS ONE. After careful consideration, we feel that it has merit but does not fully meet PLOS ONE’s publication criteria as it currently stands. Therefore, we invite you to submit a revised version of the manuscript that addresses the points raised during the review process. Your manuscript has been positively appraised by our reviewers. Notwithstanding, there are some additional revisions needed from you, as you can see below in the section "Comments to the author". Therefore, and before considering accepting this paper for publication in PLOS ONE, I must ask you to respond to all the remaining comments and suggestions in a suitable and detailed way. Once receiverd these revisions from you, I will proceed to check them personally, in order to make a prompt editorial decision on your study

We look forward to receiving your revised manuscript.

Kind regards,

Sergio A. Useche, Ph.D.

Academic Editor

PLOS ONE

Journal Requirements:

Reviewers' comments:

Reviewer's Responses to Questions

**Comments to the Author**

1. If the authors have adequately addressed your comments raised in a previous round of review and you feel that this manuscript is now acceptable for publication, you may indicate that here to bypass the “Comments to the Author” section, enter your conflict of interest statement in the “Confidential to Editor” section, and submit your "Accept" recommendation.

Reviewer #1: All comments have been addressed

Reviewer #2: (No Response)

2. Is the manuscript technically sound, and do the data support the conclusions?

Reviewer #1: Yes

Reviewer #2: Yes

3. Has the statistical analysis been performed appropriately and rigorously? 

Reviewer #1: Yes

Reviewer #2: Yes

4. Have the authors made all data underlying the findings in their manuscript fully available?

Reviewer #1: Yes

Reviewer #2: Yes

5. Is the manuscript presented in an intelligible fashion and written in standard English?

Reviewer #1: Yes

Reviewer #2: (No Response)

6. Review Comments to the Author

Reviewer #1: Thank you for your kind consideration of my questions and comments. Well done. Congratulations on a fine article.

Reviewer #2: Thank you for you revision which reads well.

One final comment - you conclude 'Significant RTA-related thoracic injuries were rare' - but I think you mean 'Major cardiac, vascular, tracheobronchial and oesophageal injuries were rare...'?

7. PLOS authors have the option to publish the peer review history of their article (what does this mean?). If published, this will include your full peer review and any attached files.

Reviewer #1: No

Reviewer #2: **Yes: **Mark Fitzgerald

---

## [Author Response · Author response to Decision Letter 1]

20 Apr 2022

We would like to thank all the reviewers for their comments.

Only one comment has been raised on point 6 “review comments to the author: One final comment - you conclude 'Significant RTA-related thoracic injuries were rare' - but I think you mean 'Major cardiac, vascular, tracheobronchial and oesophageal injuries were rare...'?”

Thank you very much,

Best regards,

Response to the reviewer: 

Thank you for this comment. This has been rephrased to be more precise. Indeed, <5% of our population sustained a moderate to lethal thoracic injury, we therefore concluded that these injuries were rare.

---

## [Editor Report · Decision Letter 2]

25 Apr 2022

Road traffic accident-related thoracic trauma: epidemiology, injury pattern, outcome, and impact on mortality – a multicenter observational study

PONE-D-22-02469R2

Dear Dr. BENHAMED,

We’re pleased to inform you that your manuscript has been judged scientifically suitable for publication and will be formally accepted for publication once it meets all outstanding technical requirements.

Kind regards,

Sergio A. Useche, Ph.D.

Academic Editor

PLOS ONE
---

## [Editor Report · Acceptance letter]

28 Apr 2022

PONE-D-22-02469R2 

Road traffic accident-related thoracic trauma: epidemiology, injury pattern, outcome, and impact on mortality – a multicenter observational study 

Dear Dr. Benhamed:

I'm pleased to inform you that your manuscript has been deemed suitable for publication in PLOS ONE. Congratulations! Your manuscript is now with our production department. 

Kind regards, 

on behalf of

Dr. Sergio A. Useche 

Academic Editor

PLOS ONE